# Immunological Imbalances Associated with Epileptic Seizures in Type 2 Diabetes Mellitus

**DOI:** 10.3390/brainsci13050732

**Published:** 2023-04-27

**Authors:** Wendy N. Phoswa, Kabelo Mokgalaboni

**Affiliations:** Department of Life and Consumer Sciences, University of South Africa (UNISA), Science Campus, Private Bag X6, Florida, Roodepoort 1710, South Africa

**Keywords:** diabetes mellitus, epilepsy, immunology, inflammasomes, type 2 diabetes mellitus

## Abstract

Purpose of the review: Type 2 diabetes mellitus (T2DM) is a global health burden that leads to an increased morbidity and mortality rate arising from microvascular and macrovascular complications. Epilepsy leads to complications that cause psychological and physical distress to patients and carers. Although these conditions are characterized by inflammation, there seems to be a lack of studies that have evaluated inflammatory markers in the presence of both conditions (T2DM and epilepsy), especially in low-middle-income countries where T2DM is epidemic. Summary findings: In this review, we describe the role of immunity in the seizure generation of T2DM. Current evidence shows an increase in the levels of biomarkers such as interleukin (IL-1β, IL-6, and IL-8), tumour necrosis factor-α (TNF-α), high mobility group box-1 (HMGB1), and toll-like receptors (TLRs) in epileptic seizures and T2DM. However, there is limited evidence to show a correlation between inflammatory markers in the central and peripheral levels of epilepsy. Conclusions: Understanding the pathophysiological mechanism behind epileptic seizures in T2DM through an investigation of immunological imbalances might improve diagnosis and further counter the risks of developing complications. This might also assist in delivering safe and effective therapies to T2DM patients affected, thus reducing morbidity and mortality by preventing or reducing associated complications. Moreover, this review also provides an overview approach on inflammatory cytokines that can be targeted when developing alternative therapies, in case these conditions coexist.

## 1. Introduction

Diabetes mellitus (DM) is a chronic metabolic disorder characterised by hyperglycaemia caused by complete or partial insufficiency of insulin secretion and insulin action [1]. There are two types of diabetes mellitus, namely, type 1 (insulin-dependent) and type 2 (non-insulin-dependent), also known as adult-onset diabetes [2,3]. T2DM is the most common and accounts for 90–95% of all cases of DM [4,5]. It is predicted that the prevalence of T2DM will continue to increase in the next twenty years, and more than 70% of patients will come from developing countries, with the majority being around the ages of 45 and 64 years [6]. A recent report by the International Diabetes Federation has shown an alarming prevalence of DM, with 537 diabetic patients worldwide [7]. T2DM is multifactorial, resulting from alterations in genetic, environmental, and behavioural risk factors [8,9,10]. People with T2DM are more susceptible to various complications, often leading to premature death. Amongst these complications are different types of seizures that occur in about 25% of patients with DM [11,12,13,14]. The precise aetiology of seizures in diabetic patients remains unclear; however, it is speculated to result from physiological factors, such as oxidative stress, immune abnormalities, microvascular lesions in the brain, an impaired brain–blood barrier (BBB), metabolic factors, and genetic mutation [11,15,16,17,18] (Figure 1). To date, there are no international criteria for the diagnosis and treatment of this condition, thus making it difficult to diagnose and treat this condition in its early stages. Nevertheless, anti-seizure medications, such as levetiracetam, carbamazepine, lamotrigine, topiramate, and zonisamide, are commonly recommended, primarily to manage seizures in diabetic patients [19]. This can reduce secondary complications that might be associated with seizures, especially in diabetes. Furthermore, existing evidence shows that anti-diabetic medications can also be beneficial in DM patients with epileptic seizures, as they can normalise blood glucose levels and prevent a further epileptic attack when patients are no longer using such drugs [11]. Despite this evidence, it is still unclear how immunological indices and inflammatory pathways contribute to the aetiology of epilepsy in diabetic patients, especially in middle-low-income countries. Therefore, our current review discusses immunological, inflammatory imbalances, implications and genetic mutations associated with epileptic seizures in T2DM. 

## 2. Pathophysiology of Epileptic Seizures

The pathophysiology of epilepsy involves the series of events that contributes to neural hyperexcitability. Some implicated mechanisms impair ion homeostasis, neurotransmitter uptake, and blood BBB [21,22,23]. The neurons in the brain interact with each other by releasing neurotransmitters through axons, thus sending neural messages. The neurotransmitters have excitatory or inhibitory effects. The action potential of a neuron depends solely on the balance between the neurotransmitters’ excitatory and inhibitory effects [24]. The increased release of excitatory glutamate upregulates N-methyl-D-aspartate (NMDA) receptors, resulting in the accumulation of calcium ions [25]. An excess of calcium ions activates neuronal nitric oxide synthase (nNOS), which increases the production of nitric oxide (NO) and neurological apoptosis due to deoxyribonucleic acid (DNA) damage associated with excessive free radicals, all of which contributes to the development and progression of epilepsy [26] (Figure 1). 

## 3. Immunological Expression in Epilepsy

According to Vezzani et al. (2016), animal models of epilepsy are characterised by the presence of peripheral immune cells [27]. Lymphocytes, in particular, have been observed in the hippocampus following status epilepticus induced by systemic pilocarpine injection [28] or intrahippocampal administration of kainic acid in mice [29]. Active brain extravasation of these cells may alter BBB permeability [28,30]. The critical question is whether this phenomenon relates to tissue hyperexcitability or neuropathology. A study by Fabene and colleagues in 2008 revealed that epileptic seizures in an epileptic mouse model induced by pilocarpine injection upregulate the expression of vascular cell adhesion molecules (VCAM-1), concomitant to an increased leukocyte arrest in the brain [28]. This mechanism is mediated by the leukocyte mucin P-selectin glycoprotein ligand-1 (PSGL-1) and leukocyte integrins alpha-4-beta-1 [28]. This team further revealed that inhibition of contact between leukocytes and vasculature by blocking antibodies or by genetically manipulating PSGL-1 genes significantly reduces seizure attacks, with the potential of alleviating the onset of epilepsy [28]. 

In contrast, in intracerebral kainic-acid-treated mice, macrophages and T-cells prevent neutrophil extravasation, delay the onset, and thus reduce the recurrence of spontaneous seizures [29]. Epilepsy is a health burden without an identifiable cause [31,32]. The pathophysiological mechanism underlying this disease has been shown to involve both innate and adaptive immune responses, such as T- and B-cell activation, auto-AB production, and activation of inflammatory responses in the epileptogenic foci [33,34]. Immune cells activating inflammatory response have been observed in brain cells, such as microglia, astrocytes, and neurons, from patients with pharmacoresistant epileptic seizures [35]. In addition to an inflammatory response in epileptic seizures, cytokines also activate inflammation in epileptic seizures [36]. The inflammatory molecules observed in epileptic brain tissues are not only responsible for an inflammatory response per se; however, they function as neuromodulators by activating their receptors expressed by neurons, thus affecting neuronal function and excitability [37]. Therefore, it is speculated that the over-activation of inflammatory response in brain cells and the neuronal regions might contribute to the pathophysiological mechanism of seizure generation [38,39]. It is consequently critical that considerable research focus be placed on understanding the implication of immunological mechanisms in the aetiology of epilepsy. A better understanding of these mechanisms in epilepsy might be explored as a target for developing alternative therapies to epileptic seizures.

## 4. Inflammation in Epilepsy and Diabetes Mellitus

### 4.1. Central Inflammation in Epilepsy

The nuclear factor kappa-beta (NF-κβ) signalling system regulates inflammation and further contributes to inflammation observed in epilepsy [40,41,42]. NF-κβ initiates neuroinflammation through a series of mechanisms, including cyclooxygenase (COX-2), the mammalian target of rapamycin, and mitogen-activated protein kinase. These mechanisms upregulate the gene expression and activity of NF-κβ [43]. Additionally, HMGB1, NF-κβ, TNF-α, and IL-1 may, respectively, activate toll-like receptor 4 (TLR-4), TNF-receptor (TNFR), and IL-1R [44]. 

Evidence from animal models of epilepsy has shown an upregulated expression of COX-2 in hippocampus neurons after one hour of seizure; however, treatment with COX-2 inhibitors exhibited beneficial effects [45,46,47]. Prostaglandin (PGE2), a derivative of an enzyme, COX-2, contributes to the activation of prostaglandin 1, 2, 3, and 4 receptors (EP1, EP2, EP3, and EP4). This further promotes the release of calcium ions and mediates activities that impair neural function, including plasticity, neurologic deficiency, and hyperexcitability [48]. Interestingly, this reduces the threshold of seizure amongst animals and individuals with epilepsy [49,50]. Although inflammation is crucial to the aetiology of epilepsy, it is critical that future clinical investigations focus on the fundamental knowledge of inflammatory pathways that appear to promote and exacerbate epileptic complications.

### 4.2. Inflammation in T2DM

In hyperglycaemic conditions, insulin resistance results in inflammation. It has been revealed by several researchers that T2DM is associated with inflammation. Our group has previously shown an increased level of tumour necrosis factor alpha (TNF-α), interleukin-6 (IL-6) and interleukin-1-beta (IL-1β) in T2DM [51]. This suggests that increased inflammation is associated with cardiovascular diseases. Notably, the anti-inflammatory cytokine IL-4 is significantly decreased amongst T2DM patients [52]. T2DM is exacerbated by an imbalance between pro-inflammatory and anti-inflammatory cytokines.

## 5. Brain Cells as a Primary Site for the Activation of Inflammatory Responses in Epileptic Seizures

### 5.1. Glial Cells

Glial cells are reported to be the primary site for inflammatory molecule activation during epileptic seizures [36,53,54,55,56]. Upon their activation, they release several pro-inflammatory cytokines, such as IL-1β and TNF-α [36,57] (Figure 2). High-mobility group box-1 (HMGB-1) has been widely recognised as a biomarker of epilepsy [35,58]. It mediates the immune response by stimulating macrophage and endothelial cell activation, which triggers the production of TNF-α, IL-1, and IL-6 by binding to the receptor for advanced glycation end products (RAGE) and Toll-Like-receptors (TLR)-4. This further activates the nuclear factor kappa-light-chain-enhancer of activated β-cells (NF-κβ), subsequently increasing pro-inflammatory cytokine levels [59]. Evidence from preclinical studies shows that animal models exposed to lipopolysaccharide (LPS)-induced inflammatory response develop seizure due to pronounced levels of pro-inflammatory cytokines, such as IL-1β, TNF-α, and HMGB1, as seen in epilepsy comorbidities [57,60,61]. The accumulation of extracellular HMGB1, following apoptosis and highly elevated pro-inflammatory cytokine levels, promotes inflammation, which exacerbates the condition. Notably, these cytokines are partly elevated in both T2DM and epilepsy (Figure 2).

### 5.2. Microglia and Astrocytes

Animal models have shown other types of brain cells (microglia and astrocytes) that were affected during the pathophysiological mechanism of epileptic seizures [62]. It has been reported that activated microglia play a principal role in the production of cytokines that are also involved in the pathophysiology of epileptic seizures [63]. Microglia are called brain-resident immune cells partly because they can produce and release various cytokines [64]. A study by Benson et al. (2015) investigated the microglial expression of inflammatory cytokines using flow cytometry and quantitative real-time polymerase chain reaction (PCR) in a pilocarpine-induced SE. Their findings indicated that the microglia increased the expression of pro-inflammatory cytokines (IL-1β, IL-6, TNF-α, Arg1, IL-4, and IL-10) [65]. Cytokine activation in the microglial cells of people with epilepsy has been shown to occur due to TLR signalling. Microglial cells respond to the TLR3 agonist polyinosinic and the TLR4 agonist lipopolysaccharide (LPS), thus promoting the production of inflammatory cytokines [66]. The TLR has also been implicated in the pathophysiology of diabetes mellitus. Kolek et al. (2004) showed that TRL4 [lipopolysaccharide (LPS) receptor] affects the innate immune response as well as the prevalence of T2D or metabolic syndrome and atherosclerosis [67]. TRL4 has also been reported to promote insulin resistance [68]. Insulin resistance contributes to the pathogenesis of DM, more particularly T2DM [69]. Interestingly, the findings by Wang et al. (2015) demonstrated the hyperglycaemia-induced overexpression and activation of TLR4 in endothelial cells. They further revealed that TLR4 leads to the activation of inflammatory responses that contribute to the pathogenesis of diabetic retinopathy [63]. More interestingly, Taha et al. (2018) suggested that high levels of TLR4 are associated with T2DM [70].

TLR4 plays a pivotal role in the brain and central nervous system (CNS) by regulating neuroinflammation [71]. The activation of TLR4 has a beneficial scavenging effect on amyloid beta (Aβ). However, the chronic activation of Aβ leads to Aβ deposition in the brain and has been widely associated with the pathogenesis of Alzheimer’s disease [72]. Alzheimer’s disease is a neurodegenerative disease that is associated with complications of diabetes mellitus [73]. TLR4 has also been reported to link diabetes mellitus and Alzheimer’s disease [74]. Therefore, we speculate that the TLR4 signalling pathway may also be a potential link between T2DM and epileptic seizures. TNF-α is released from activated microglia and astrocytes, and it can induce epilepsy. In a nutshell, TNF-α controls the synaptic function in astrocytes and regulates brain activity by increasing glutamate release, decreasing the production of gamma-aminobutyric acid, generating neuroinflammation [75]. Therefore, any therapeutic approach that can reduce TNF-α-associated glutamate may be of relevance in the search for an ideal epileptic treatment.

Microglial stimulation has been demonstrated to contribute to epilepsy through the HMGB1-TLR2/4-NF-kβ-mediated pathway. Of interest is the unique potential of this protein, HMGB1, as a treatment and non-invasive biomarker for epilepsy and patients at high risk of developing epilepsy [76]. It has been proven that the HMGB1 level rises after four hours of drug-resistant epilepsy seizure events, making HMGB1 a promising marker for epilepsy [77]. On the other hand, children with febrile seizures have increased serum levels of HMGB1 compared to normal children [78]. An increased level of HMGB1 induces inflammation, which may lead to additional complications associated with epilepsy. It is therefore critical to reduce inflammation in epileptic patients in order to reduce inflammatory-related secondary effects. Recently, the therapeutic strategy for reducing HMGB1 has been re-investigated, and the use of HMGB1 inhibitors has been well described in mouse models, with encouraging outcomes [79].

## 6. Peripheral Inflammatory Cytokines in Epilepsy

Brain inflammation is not the only source of the pathogenesis of epilepsy; peripheral inflammation also plays a critical role in the development of epilepsy [80]. Several findings have indicated that central-nervous-system inflammatory cytokines correlate with peripheral inflammation. A study by Chmielewska et al. (2021) observed increased plasma IL-1β and IL-6 in electrically induced hippocampal epilepsy in rats [81]. In agreement with this, Huang et al. (2018) also reported increased serum IL-2 and IL-6 in a rat model of seizures induced by intraperitoneal injection with kainic acid [82]. More recently, their findings were validated by Shin et al. (2022), who also observed increased serum IL-2 and IL-6 in tonic-clonic seizures [83]. 

Interestingly in epileptic patients, Basnyat et al. (2023) reported elevated IL-6 levels in the blood, which were associated with high antibodies against glutamic acid decarboxylase (GADA) titters [84]. These suggest that IL-6 can be used as a biomarker to understand the immunological pathways implicated in the pathophysiology of GADA-associated autoimmune epilepsy. HMGB1 is an inflammatory marker widely associated with epilepsy; however, currently, there is a paucity of evidence showing a correlation between central and peripheral HMGB1 levels. For example, Wang et al. (2021) determined the expression profiles of HMGB1 in cerebrospinal fluid (CSF) and paired serum [85]. They found that HMGB1 was only increased in the CSF, without a correlation between CSF and serum HMGB1 levels [85]. They further concluded that HMGB1 might be the main contributor to seizure mechanisms and that CSF HMGB1 can be used as a predictive marker in epilepsy [85]. Moreover, an understanding of the epileptic neurobiology of inflammation might be crucial to the identification of ideal markers that can be used as alternative therapeutic targets for the prevention and treatment of epileptic seizures.

## 7. Immunological Factors Commonly Contribute to the Pathogenesis of Diabetes

Immunological factors that commonly contribute to the pathogenesis of DM have been associated with two factors: (1) the activation of inflammasomes and (2) the release of pro-inflammatory cytokines in response to damage-associated molecular patterns (DAMPs). The inflammasome is a multiprotein complex needed for caspase-1 processing and activating inflammatory cytokines, such as IL-1β and IL-18 [86]. This activation of inflammatory cytokines occurs when damage-associated DAMPs, pathogen-associated molecular pattern molecules (PAMs, e.g., lipopolysaccharide), nucleotide-binding oligomerisation domain-like receptors (NLRs) or absent in melanoma 2 (AIM2) form a protein complex with pro-caspase-1 and the activation of TLR [86,87].

The aberrant activation of inflammasomes has been associated with the pathogenesis of autoimmune, autoinflammatory, chronically inflammatory, and metabolic diseases [88,89]. The NLR family activates abundant inflammasomes [90], which have been characterised by (1) the NLRP1/NALP1b inflammasome [91], (2) the NLRC4/IPAF inflammasome [92], (3) the NLRP3/NALP3 inflammasome [93], and (4) the AIM2-containing inflammasome [94]. Interestingly, NLRP3 inflammasomes are said to be activated by both exogenous (including DAMPs and MAPs) [95] and endogenous molecules, such as crystalline molecules, extracellular ATP, receptor P2X7 receptor (P2X7R) (through its cell surface), A fibrils, lipopolysaccharide (LPS), hyaluronan, and uric acid crystals [96].

Moreover, ATP receptors, such as P2X4, P2X7, P2Y4, P2Y6, P2Y7, and P2Y12, have been reported to be expressed by microglial cells [97,98]. Avignone et al. (2018) reported that P2Y6, P2Y4, P2Y6, P2Y7, and P2Y12 mRNA levels increased in the hippocampus in a kainic-acid-induced mouse model of mesial temporal lobe epilepsy (MTLE) after status epilepticus [99]. It is known that P2X7 activation promotes IL-1β processing and TNF-α expression, which are implicated in the pathophysiology of seizures [100]. However, regarding whether P2X7 has proconvulsive or anticonvulsive effects that vary according to the animal models of status epilepticus, P2X7 plays proconvulsive roles in pilocarpine-induced status epilepticus and anticonvulsive roles in status epilepticus triggered by intra-amygdala injection of kainic acid [101,102]. In addition to the role of NLRs, TLRs are implicated in the pathogenesis of T2DM and related complications [103,104,105,106]. Hence, this suggests that TLRs contribute to the disease’s development and to the pathogenesis of this condition [87]. It has been documented that TLR increases blood glucose levels, pro-inflammatory cytokines, and oxidative stress [107,108] (Figure 1). 

There is increasing evidence linking HMGB1 with T2DM and obesity. A study conducted in China reported that increased plasma HMGB1 was associated with insulin resistance, whereas increased serum HMGB1 was linked to pro-inflammatory cytokines induced by T2DM and obesity [109]. Jeong et al. (2022) recently indicated that HMGB1 could exacerbate brain insulin resistance by reducing insulin receptor expression and deactivating the insulin signalling pathway. This can negatively impact the brain cells by preventing glucose transportation to these cells, thus leading to epileptic seizures [110]. In addition, T2DM can lead to oxidative stress, which causes BBB damage [111,112,113]. Damaged BBB can be vulnerable to neurological conditions such as epilepsy [114,115]. Notably, BBB leakage induces epilepsy [116] by increasing the production of glutamate. An increase in glutamate promotes excitatory activity in the brain, thus resulting in the development and exacerbation of epileptic seizure [117]. Recently, Chen and colleagues have reported the use of glutamate receptor antagonists as an effective treatment against epilepsy [117].

Other cytokines that are reportedly increased in T2DM include IL-6 [118,119,120,121,122], IL-2 [123,124], and TNF-α, which have been strongly associated with the pathogenesis of T2DM [125,126]. A vast trove of evidence from previous literature indicates similarities in immunological changes that occur in diabetes and epilepsy, which explains why T2DM patients are at an increased risk of being predisposed to epileptic seizures or epilepsy in the long run. In addition, both T2DM and epileptic aetiology seems to emanate from similar inflammatory pathways, as demonstrated by increased pro-inflammatory cytokines. This suggests an alternative approach that researchers can focus on when developing therapies against complications in both T2DM and epilepsy. 

## 8. Conclusions

The aetiology of epilepsy is multifaceted; however, the imbalance in anti- and pro-inflammatory cytokines seems to implicate pathways in CNS and systemic tissue. The available evidence from preclinical and clinical studies shows similarities in the expression of immunological, pro-inflammatory and anti-inflammatory biomarkers, including IL-1β, IL-6, IL-8, HMGB1, TNF-α, TLRs, and PX27, implicated in the pathogenesis of T2DM and epileptic seizures. This suggests that there could be a close correlation between these two conditions. This creates a background that future research can focus on to further develop therapies against inflammatory conditions in instances where these two conditions coexist. Some of the approaches may include targeting HMGB1, which seems to exacerbate inflammation in epilepsy. Monitoring alterations in the BBB structure is also recommended in T2DM in order to prevent the pathogenesis of epilepsy. Despite this evidence, it is still crucial to conduct more clinical studies to ascertain these finding.

## Figures and Tables

**Figure 1 brainsci-13-00732-f001:**
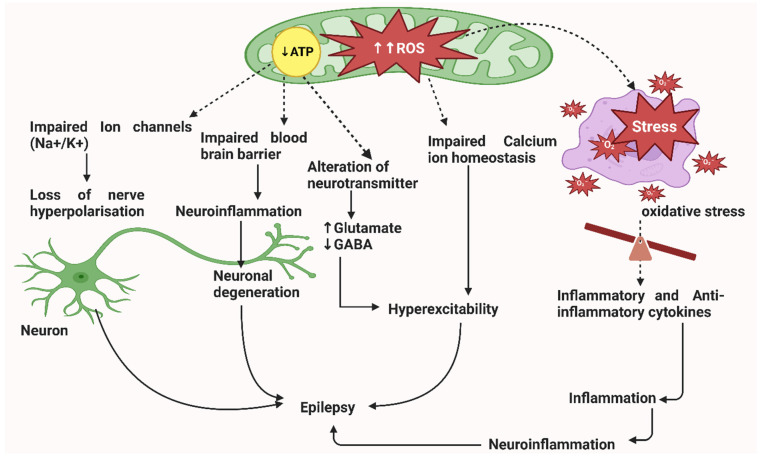
Brief mechanism of the pathophysiology of epilepsy. Different pathways can induce this; for example, oxidative stress due to an increased circulating reactive oxygen species (ROS), which results in imbalanced inflammatory cytokines. The latter stimulates inflammation and subsequently leads to neuroinflammation. Similarly, ROS can impair calcium ion metabolism, resulting in neural hyperexcitability. The loss of mitochondrial adenosine-triphosphate (ATP) has also been implicated in the pathogenesis of epilepsy. This seems to occur in three different pathways; for instance, the alteration of neurotransmitters due to the lack of ATP can induce neural hyperexcitability by increasing glutamate and the reduction of gamma-aminobutyric acid (GABA) in a mitochondrion. Secondly, less ATP impairs the sodium and potassium ion channels, resulting in nerve depolarisation. On the other hand, the impaired brain–blood barrier (BBB) also triggers neuroinflammation, causing neural degeneration. Altogether, these mechanisms cause neuronal degeneration, which facilitates the pathogenicity of epilepsy [20].

**Figure 2 brainsci-13-00732-f002:**
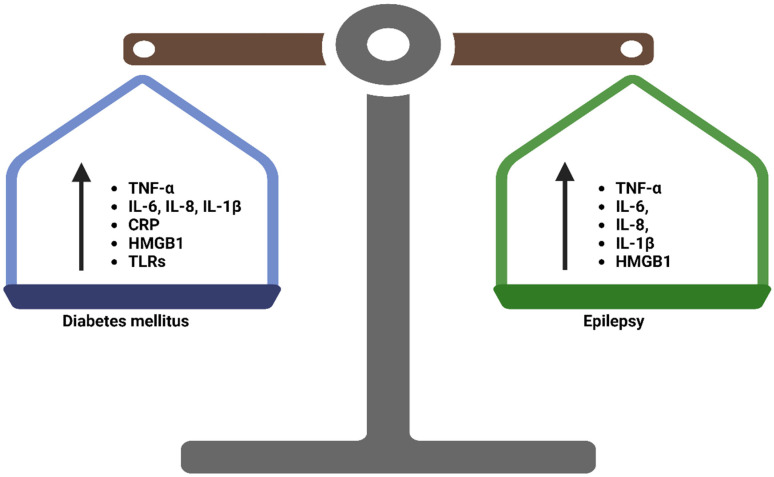
Demonstration of the level of circulating pro-inflammatory cytokines in epileptic seizures and T2DM. In both conditions, there is an increased level of pro-inflammatory cytokines, implying that there could be a direct link or inflammatory relationship between them. TNF-α, tumour necrosis factor-alpha; IL, interleukin; CRP, C-Reactive Protein; HMGB1, High-mobility group box-1; TLRs, toll-like receptors.

## Data Availability

Data sharing is not applicable to this article as no datasets were generated or analysed during the current study.

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
