# Peer review of "Immunological Imbalances Associated with Epileptic Seizures in Type 2 Diabetes Mellitus"

_brainsci, 2023, doi:10.3390/brainsci13050732_

Round 1

Reviewer 1 Report

Thank you for inviting me to review this manuscript. 

This review aimed to review the role of immunity in seizure generation in T2DM patients.

In the beginning,  the authors should distinguish between "acute symptomatic seizures" in T2DM patients related to hyperglycemia and "epilepsy in T2DM patients". In this review, did the author tries to explain the occurrence of acute symptomatic seizures associated with hyperglycemia or the occurrence of epilepsy in T2DM patients?

The authors explained the role of pro-inflammatory factors in epilepsy and diabetes mellitus well. In animal models, many pro-inflammatory factors were found in the hippocampus, glial cells, microglia, and astrocytes. However, this review has not explained a clear relationship between increased pro-inflammatory factors in the blood in T2DM patients and increased pro-inflammatory factors in the CNS. There is a blood-brain barrier that protects the brain against inflammation occurring in the blood. Pro-inflammatory factors in the blood can be correlated with increased pro-inflammatory factors in the brain when there is damage to the blood-brain barrier. The author should explain this. Does blood-brain barrier damage occur in T2DM patients?

The terminology of anti-epileptic drugs has been replaced by anti-seizure medication nowadays. 

Line 39: “Anti-epileptic drugs such as diazepam and carbamazepine are commonly recommended in diabetic patients [20,21]”. The authors referred to >10 years of references. Diazepam is used for acute seizures and carbamazepine is used as maintenance anti-seizure medication (ASM) for epilepsy. It should be distinguished between those drugs. Seizure type in T2DM is commonly focal onset seizure and the first line ASM should be lamotrigine or levetiracetam based on NICE guideline 2022. 

There are some mistyped in the manuscript.

Please use references < 10 years. 

Author Response

  1. Comment 1: In this review, did the author tries to explain the occurrence of acute symptomatic seizures associated with hyperglycemia or the occurrence of epilepsy in T2DM patients?

Response 1: In this review the authors are reporting on the association between seizures and T2DM.

  1. Comment 2: This review has not explained a clear relationship between increased pro-inflammatory factors in the blood in T2DM patients and increased pro-inflammatory factors in the CNS. 

Response 2: The authors had added a section indicating a correlation between central and peripheral cytokines in epilepsy and peripheral cytokines in T2DM. The sections are highlighted in green on the manuscript.

  1. Comment 3: Does blood-brain barrier damage occur in T2DM patients?

Response: The authors would like to thank the reviewer for this question. The answer is yes, and this information has been added on the manuscript as highlighted in blue.

  1. Comment 4: The terminology of anti-epileptic drugs has been replaced by anti-seizure medication nowadays. Line 39: “Anti-epileptic drugs such as diazepam and carbamazepine are commonly recommended in diabetic patients [20,21]”.

Response 4: The Authors appreciate the comment made by the reviewer. The term anti-epileptic drugs have been replaced with anti-seizure medication as suggested. Changes are highlighted in yellow.

  1. Comment 5: The authors referred to >10 years of references.

Response 5: The authors have added more references that are <10 years.

  1. Comment 6: There are some mistyped in the manuscript.

Response 6: The authors have corrected mistypes in the manuscript as highlighted in yellow.

Reviewer 2 Report

Overall, it is well organized, but the following items need to be supplemented.

I think it is necessary to classify and organize the discussion on the pathophysiology of epileptic seizures by major factors.

The following part is a core part of this study, and it is thought that there is a need to add more detailed discussion by classifying each factor.

Immunological expression in epilepsy

Immunological factors commonly contribute to the pathogenesis of diabetes

Author Response

Comment 1: The following part is a core part of this study, and it is thought that there is a need to add more detailed discussion by classifying each factor.

Immunological expression in epilepsy

Immunological factors commonly contribute to the pathogenesis of diabetes.

Response: Thank you very much for this comment which has improved our manuscript. More detailed discussion has been added on the manuscript as highlighted in green.

Round 2

Reviewer 1 Report

Thank you for the revised manuscript. 

Line 39: “Anti-seizure medications such as diazepam and carbamazepine are commonly recommended primarily to manage seizures in diabetic patients [20,21]”. The authors referred to >10 years of references. Diazepam is used for acute seizures and carbamazepine is used as maintenance anti-seizure medication (ASM) for epilepsy. It should bed distinguished between those drugs. Seizure type in T2DM is commonly focal onset seizure and the first line ASM should be lamotrigine or levetiracetam based on NICE guideline 2022. 

Line 102: “Peripheral Inflammatory cytokines.” There is only the subheading title in this line. 

The authors explained the role of peripheral inflammatory cytokines in epilepsy and added that oxidative stress in T2DM patients could lead to BBB damage. However, the reference to this statement (Reff no. 6) was not correlated with this statement. The author should refer to the original article that discusses this. 

The authors had some references that were < 10 years. But the numbering of the citation should be revised from the beginning. 

Please use the references <10 years and replace those >10 years' references. 

Author Response

We would like to thank you for reviewing our manuscript and  for all the comments made to improve the manuscript. All comments have been addressed and changes to the manuscript have been highlighted in yellow.

REVIEWER 1

  1. Comment 1: Line 39: “Anti-seizure medications such as diazepam and carbamazepine are commonly recommended primarily to manage seizures in diabetic patients [20,21]”. The authors referred to >10 years of references. 

Response: The authors have updated the reference and included a more recent citation.

  1. Comment 2: Diazepam is used for acute seizures and carbamazepine is used as maintenance anti-seizure medication (ASM) for epilepsy. It should bed distinguished between those drugs. Seizure type in T2DM is commonly focal onset seizure and the first line ASM should be lamotrigine or levetiracetam based on NICE guideline 2022. 

Response: According to NICE guideline 2022 loreclezole and carbamazepine were the two ASMs which had the highest point estimate of relative effectiveness. Effectiveness for cenobamate was also greater than that for lamotrigine, levetiracetam and gabapentin all drugs recommended in the previous NICE guideline for add-on treatment for focal seizures.

  1. Comment 3: Line 102: “Peripheral Inflammatory cytokines.” There is only the subheading title in this line. 

Response: The subheading has been move to section number 6 of the manuscript.

  1. Comment 4: The authors explained the role of peripheral inflammatory cytokines in epilepsy and added that oxidative stress in T2DM patients could lead to BBB damage. However, the reference to this statement (Reff no. 6) was not correlated with this statement. The author should refer to the original article that discusses this. 

Response: The authors would like to thank the reviewer for this comment. Reference 6 has been removed and replaced with new references as highlighted in blue.

  1. Comment 5: The authors had some references that were < 10 years. But the numbering of the citation should be revised from the beginning. 

Response: The numbering of the citation has been updated.

  1. Comment 6: Please use the references <10 years and replace those >10 years' references. 

Response: The authors have updated most references on the manuscript. However, in some instances we could only find references that are >10 years.

Reviewer 2 Report

You are a good jobs.

Entirely, this paper was revised well for publication.

Author Response

The authors would like to appreciate the reviewer for their review comments which have improved the quality of the manuscript.